# A Novel Automatic Audiometric System Design Based on Machine Learning Methods Using the Brain’s Electrical Activity Signals

**DOI:** 10.3390/diagnostics13030575

**Published:** 2023-02-03

**Authors:** Mustafa Küçükakarsu, Ahmet Reşit Kavsaoğlu, Fayadh Alenezi, Adi Alhudhaif, Raghad Alwadie, Kemal Polat

**Affiliations:** 1Department of Biomedical Engineering, Faculty of Engineering, Karabuk University, Karabuk 78050, Turkey; 2Department of Electrical Engineering, Jouf University, Sakaka 72388, Saudi Arabia; 3Department of Computer Science, College of Computer Engineering and Sciences in Al-Kharj, Prince Sattam Bin Abdulaziz University, P.O. Box 151, Al-Kharj 11942, Saudi Arabia; 4General Directorate of Health Affairs, Asir Region, Al Rabwah, 2712, 8037, Abha 62523, Saudi Arabia; 5Department of Electrical and Electronics Engineering, Faculty of Engineering, Bolu Abant Izzet Baysal University, Bolu 14280, Turkey

**Keywords:** audiometry, brain signals, EEG, machine learning, automatic audiometric system

## Abstract

This study uses machine learning to perform the hearing test (audiometry) processes autonomously with EEG signals. Sounds with different amplitudes and wavelengths given to the person tested in standard hearing tests are assigned randomly with the interface designed with MATLAB GUI. The person stated that he heard the random size sounds he listened to with headphones but did not take action if he did not hear them. Simultaneously, EEG (electro-encephalography) signals were followed, and the waves created in the brain by the sounds that the person attended and did not hear were recorded. EEG data generated at the end of the test were pre-processed, and then feature extraction was performed. The heard and unheard information received from the MATLAB interface was combined with the EEG signals, and it was determined which sounds the person heard and which they did not hear. During the waiting period between the sounds given via the interface, no sound was given to the person. Therefore, these times are marked as not heard in EEG signals. In this study, brain signals were measured with Brain Products Vamp 16 EEG device, and then EEG raw data were created using the Brain Vision Recorder program and MATLAB. After the data set was created from the signal data produced by the heard and unheard sounds in the brain, machine learning processes were carried out with the PYTHON programming language. The raw data created with MATLAB was taken with the Python programming language, and after the pre-processing steps were completed, machine learning methods were applied to the classification algorithms. Each raw EEG data has been detected by the Count Vectorizer method. The importance of each EEG signal in all EEG data has been calculated using the TF-IDF (Term Frequency-Inverse Document Frequency) method. The obtained dataset has been classified according to whether people can hear the sound. Naïve Bayes, Light Gradient Strengthening Machine (LGBM), support vector machine (SVM), decision tree, k-NN, logistic regression, and random forest classifier algorithms have been applied in the analysis. The algorithms selected in our study were preferred because they showed superior performance in ML and succeeded in analyzing EEG signals. Selected classification algorithms also have features of being used online. Naïve Bayes, Light Gradient Strengthening Machine (LGBM), support vector machine (SVM), decision tree, k-NN, logistic regression, and random forest classifier algorithms were used. In the analysis of EEG signals, Light Gradient Strengthening Machine (LGBM) was obtained as the best method. It was determined that the most successful algorithm in prediction was the prediction of the LGBM classification algorithm, with a success rate of 84%. This study has revealed that hearing tests can also be performed using brain waves detected by an EEG device. Although a completely independent hearing test can be created, an audiologist or doctor may be needed to evaluate the results.

## 1. Introduction

Hearing loss has become a chronic disease in people of all ages. According to the World Health Organization, 466 million people worldwide have hearing loss, and 75% are in developing countries [1]. Although hearing impairment is not fatal, it can cause depression, communication problems, and functional problems in daily life [2]. According to the research, it has been seen that people who have hearing loss and then use hearing aids benefit more than those who are diagnosed in the later stages and start using the device [3]. For this reason, the hearing test should be done without neglecting it. Pure tone audiometry testing is the basis of hearing assessment [4].

In the current systems, hearing tests are performed by the audiologist in a soundproof cabinet in a clinic. During the trial, pure tones are given to the person at different levels (250 Hz, 500 Hz, 1000 Hz, 2000 Hz, 4000 Hz, 8000 Hz) with intensities between 0 dB and 110 dB through insulated headphones [5]. If the individual states that they hear the given sound, the sound intensity is reduced by 10 dB, while if they do not state that they hear the given sound, it is increased by 5 dB. This hearing test procedure is called the Hughson-Westlake method [6]. When sounds are given during the hearing test, it will influence the brain if they are heard. When the sound is not given, and the sounds provided are not heard, an impact on the brain will not be expected. In this study, signals are assigned randomly to create an effect in the brain with the designed interface. Sounds are randomly given to the person in terms of amplitude and frequency. If the person heard, Considering that he will listen to higher wavelengths of that frequency, those sounds were not given, but lower-intensity sounds were given. If the person has not heard, it is not shown in the frequency of the sound that he has not heard, considering that he will not hear lower intensity, but the higher intensity sounds are given. 

In this paper, the system whose block diagram is shown in Figure 1 is designed. The hearing test user interface was prepared in the MATLAB GUI environment, and the pre-processing of the raw EEG data and the execution of the machine learning algorithms were performed with the PYTHON programming language. With the hearing test interface, sound signals were randomly given to the person being tested, and the person was expected to press a mouse button or a key on the keyboard to the sounds they heard. Brain Products’ V-Amp 16 model product was used to receive and process brain signals in real-time, and thus our EEG data were studied.

The rest of this study is organized as follows: the material and the proposed method are given in Section 3. Then, the experimental results and discussion are presented in Section 4. Lastly, concluding remarks are provided in Section 5.

## 2. Literature Review

Gargouri M et al., developed a portable system to perform individual hearing testing. They used the Raspberry Pi3 B+ board to generate the audio signal and evaluate the test results. They performed the hearing test by connecting a headset and mouse to this card, which provides computer features. While testing the right ear, no sound was given to the left ear, and no sound was given to the right ear when trying the left ear. Initially, they gave a sound signal of 40 dB and 1000 Hz, decreased the decibel value in 20 dB steps until the person did not indicate that they heard it, and then increased the decibel value in 10 dB steps until they stated that they heard it. They completed the hearing test according to the signals heard by the person [7].

In another paper, Ilay Saka performed the hearing test with a ready-made mobile application and a hearing impairment scale. The hearing impairment scale consists of 25 items, and there are “Yes”, “No”, and “Sometimes” options as answers. Zero points are given for each “No” answer, two points for the “Sometimes” answer, and four points for the “Yes” answer. When all the scores are added together the results are: “no obstacle” for 0–17, “mild-moderate dis-ability” for 18–42, and “significant disability” for values greater than 43, in his study with the mobile application, individuals were given sounds in different dB and frequencies. The person was expected to indicate whether they heard or not. Ilkay Saka, in his study, determined that the use of the hearing aid had no effect on the results of the tests and stated that the self-made hearing test applications could be applied independently of the use of the hearing aid [8].

In another paper, Sadık Özçelik et al. showed the effect of music education on musical hearing and perception with statistical measurements and neural network analysis. The study applied audiometric, single-tone vertical hearing, multi-tone horizontal hearing, melody, and rhythm hearing tests to the students entering the Faculty of Education. The students were subjected to the same tests at the end of two years, and the prediction of the music department and the students of the other departments was carried out with the neural network method. They showed that they classified the music department students with a success rate of 92% and the students of other departments with a success rate of 88% [9].

The study by Rajkumar S. and colleagues used artificial neural network methods to determine the hearing loss value and the appropriate hearing aid gain value. It was reported that 86% of individuals with hearing impairments were satisfied with the gain values recommended by the studies [10].

In their study, Franz Fürbass and colleagues developed an algorithm using deep learning and could accurately predict whether an individual had epilepsy or not with 80% success, based on EEG data [11].

Su-Lim Tan and colleagues conducted a study that combined pure tone audiometry and speech recognition tests and designed it with a microcontroller. The speech recognition test measures an individual’s ability to understand speech in a noisy environment. Audiologists usually conduct this test by having the person speak into a microphone or by playing pre-recorded speeches from a recording environment. In the study carried out by Su-Lim Tan and colleagues, they developed a system that independently performs these two tests without the help of audiologists [12].

In the study by Ykhlef Fayçal and colleagues, they created a hearing test system using pure tone sound and white noise masking signal with the help of a computer with a standard sound card. White noise is a broadband noise with the same acoustic energy at all frequencies. White noise has the feature of masking distracting, disturbing, and focus-preventing sounds. The study developed a system to produce similar sounds to those produced by an audiometer device and detect patients’ hearing loss thresholds [13].

In their study, Muhammad Yeamim Hossain and his colleague created an artificial intelligence model using the SVM (Support Vector Machine) algorithm, a machine learning classification method for detecting human motor-neuron behavior using EEG signals [14].

In his study, N. Sriraam presented a study that automatically detects hearing loss using EEG signals. The study tested the effect of auditory stimuli on newborn babies’ brains. Advanced feedforward and feedback neural network models were used on the collected EEG data. The studies were conducted using Multilayer Perceptron Neural Network (MLP) and Elman Neural Network (EN) artificial neural network classification algorithms. Even though the learning time of the EN algorithm was longer than that of the MLP method, it was observed that the classification prediction accuracy was higher [15].

In the study by Paulraj M P and colleagues, the responses of newborn babies’ brains to different decibels and frequencies of sounds were investigated using EEG. Studies using MLP and EN, classification algorithms of artificial neural networks, resulted in a detection accuracy rate of 79.99% for the left ear and 82.78% for the right ear [16].

In their study, Priyatam Naravajhula and colleagues carried out a spam classification prediction study using the TF-IDF method on SMS and Twitter text data with Artificial Intelligence. The most successful algorithm was Logistic regression, with a 95.3% Accuracy value [17].

## 3. Materials and Methods

### 3.1. Material

To create a stimulus in the brain, sound signals of different frequencies and decibels are given to the person tested on the designed MATLAB GUI via headphones. For the sounds heard, the person pressed a button with the computer’s mouse or a key on the keyboard. The person did not act during the waiting periods between inaudible sounds and two different sounds. To follow the brain waves during this hearing test, electrodes were positioned on the skin, and the brain waves were recorded. After the EEG signals were separated as audible and inaudible signals, they were subjected to pre-processing studies for machine learning. Then, training and testing processes were carried out with classification algorithms. For hearing tests to be carried out, they must be given to the person who produces the sounds in the desired format and is tested appropriately. Software and sound card hardware are needed to make the sounds to be used in the hearing test, and headphones are required to transmit them.

A 16-channel EEG device was utilized in this study. Data collected from volunteers displayed the P300 wave upon visualization. In addition to the 16 channels, 13 and 8 channels where the P300 effect was more substantial were also utilized as training data. The positions of these channels were sourced from various studies. Figure 2 shows the locations of the EEG electrodes used on the scalp.

#### The MATLAB GUI Hearing Test Interface and EEG Signals Recorded during the Process

Hearing tests have an important place in the detection of auditory disease. Today, hearing tests are performed by an audiologist in a clinical setting, in sound-isolated rooms. The on-duty audiologist gives the person to be tested a headset and a button to indicate that they have heard the sounds. The audiologist provides sound signals with frequency values of 125 Hz, 250 Hz, 500 Hz, 1 kHz, 2 kHz, 4 kHz, and 8 kHz, with intensity varying between 0 dB–110 dB. If the patient states that he has heard, the power is reduced by 10 dB for the same frequency sound, and a response is expected again. If the patient does not state that he hears, that is, he does not press the button, the sound intensity is increased by 5 dB for the same frequency value. This hearing test method is the Hughson Westlake Ascending Method [5]. In this study, to be able to perceive the stimuli clearly in the brain, the frequencies of 125 Hz, 250 Hz, 500 Hz, 1 kHz, 2 kHz, 4 kHz, 8 kHz, and 12 kHz and 10 dB, 20 dB, 30 dB, 40 dB, 50 dB, 60 dB. Sounds with the intensity of 70 dB, 80 dB, 90 dB, 100 dB, 110 dB, and 120 dB are given randomly. Random frequency and intensity matching was made, and this test was given to the person tested. Since there are eight different frequency values and 12 different decibel values, 96 different sounds must be given to the person to complete the test for one ear. The total number of sounds required for both ears will be 192. It has given too many sounds that can take a long time and causes the person to make an error.

For this reason, if the voice given to the person is heard by the person, it was accepted that they would hear a similar frequency and higher decibels, and the same sounds were not given to the person again. Likewise, for the sounds that the person cannot hear, sounds with the same frequency but lower decibels cannot be heard, so the sounds in this decibel are not given to the person again. The hearing test interface is shown in Figure 3.

The person to be hearing the test was put on headphones and given a keyboard or mouse to indicate the sounds he heard. If “Random db” and “Random freq” sections are selected on the interface, the sounds will be given at random frequency and intensity. If not selected, sounds in sequential frequency and decibel values will be given in the form of a standard hearing test. In this study, it was thought that sounds with random frequency and decibel value would create a more pronounced stimulus in the brain. With the “Left Ear” option, the hearing test is performed only on the left ear, with the “Right Ear” option only on the right ear, and with the “Both of Ears” option, during which the two ears are tested sequentially, first the left ear and then the right ear. The test starts with the selected settings by clicking the “START TEST” button. In this study, the left and right ears were tested sequentially with random frequency and decibel values by choosing the “Both of Ears”, “Random db” and “Random freq” options.

The test started with the left ear first, and the person clicked on the “HEAR” button or pressed a key on the keyboard after the sounds they heard. They did not act during the waiting times between the sounds they did not hear and the sounds that were given. After the first sound is given to the person, the EEG signals are recorded with the Recorder software. The S1 label is marked on the EEG signal when a sound is given to the left ear. As soon as the person stated that he heard that sound, the S2 label was marked on the EEG signal. For the right ear, the S3 label is marked when the sound is given, and the S4 label is marked when it is heard. In the middle of the pause between the two sounds, the S5 label is marked.

Figure 4 shows the test performed for both ears. If the “Both of Ears” option shown in Figure 4 is selected, the test is performed for both ears; and the left ear is tested first, then the right ear is tested. In the test performed for the left ear, the “x” sign is shown in blue for the sounds heard, and the “X” sign is shown in blue for the sounds that cannot be heard. After the left ear test is completed, the next right ear test is started. In the test performed for the right ear, the “o” sign is shown in red for the sounds heard, and the “O” sign is shown in red for the sounds that cannot be heard. If the test is requested to be repeated or a new examination is performed, the “REFRESH” button can be clicked on the screen shown in Figure 3. This hearing test can also be performed for one ear with the “Left Ear” or “Right Ear” options shown in Figure 3.

During the hearing test performed with the hearing test interface, EEG recordings were taken using the Brain Products V-Amp 16 EEG recorder (V-Amp, Brain Products GmbH, Gilching, Germany) and the Brain Products ActiCap Xpress model dry electrode. After the EEG recorder hardware is attached to the person’s head, the adjustments in Figure 5 are made on the Recorder software. According to these settings, 16 channels of EEG data will be recorded with a sampling frequency of 2000 Hz per second. EEG data recorded with the Recorder software can be filtered. In this study, a 50 Hz Notch filter and 1–12 Hz band-pass filtering was applied.

Data was collected from nine volunteers with different characteristics for the study. The volunteers consist of seven males and two females. The oldest male volunteer is 44 years old, the youngest male volunteer is 28 years old, and the average age of the male volunteers is calculated to be 35.14. The oldest female volunteer is 61 years old, the youngest female volunteer is 38 years old, and the average age of the female volunteers is calculated to be 49.5. Therefore, the average age of the nine individuals is calculated to be 38.33. Each volunteer works in different jobs. Ethical board approval was obtained to collect data from the volunteers. 

### 3.2. The Proposed Method

Figure 6 shows the Schematic representation of the proposed method. The proposed method consists of five steps:Step 1With the hearing test interface, the hearing test is performed first on the left ear and then on the right ear when the whole test is finished. With the hearing test, sound signals with frequency values of 125 Hz, 250 Hz, 500 Hz, 1 kHz, 2 kHz, 4 kHz, 8 kHz, and 12 kHz are given to the person, with an intensity varying between 0 dB–110 dB.Step 2For each given sound, the person presses the button if he/she hears it. If the button is not pressed, it is considered not heard. Brain signals are received simultaneously with EEG signals. Thus, the effect of the heard sound on the brain is determined. This marked EEG data is RAW EEG DATA.Step 3Pre-processing Steps: The following procedures were applied to the pure EEG data.50 Hz Notch Filter1–12 Hz Band-pass filterNormalization ProcessRound to 3 digits after the commaCount VectorizerTFIDF

The data was filtered using a band-pass filter (BPF) to eliminate noise and irrelevant information from low-frequency bands (1 Hz) and high-frequency bands (12 Hz) [19]. The EEG data is band-pass filtered with a bandwidth of 1–12 Hz, a conventional bandwidth for P300 detection [20], and removes noise while preserving P300 information [21].

Step 4After pre-processing, the dataset has become suitable for machine learning classification algorithms. After the entire data set was shuffled, it was split into 70% training data and 30% test data. Naïve Bayes, LGBM, SVC, DTC, KNC, L.R., and RFC algorithms were applied to the training data.Step 5Each model made an estimation on the test data, and the estimation results were compared with the actual results. For model success, K-fold cross-validation, confusion matrix, accuracy rate, R2 score, MSE, recall, precision rate, specificity rate), F1 score, MCC, and log-loss (log loss) methods were used.

#### 3.2.1. The Data Pre-Processing Method

In many machine learning algorithms, statistical models always work with vectors and matrices. To use real-world data as input to machine learning algorithms, it is necessary to convert it into a suitable format. This study has used the Count Vectorizer and TFIDF (Term Frequency Inverse Document Frequency) method.

Count Vectorizer:

In this study, EEG signals were normalized and converted to text format. For the data converted to text format to be run in machine learning algorithms, it is necessary to vectorize it. Count Vectorizer converts a text into a vector depending on the frequency of each word in the entire text [22].

Table 1 shows eight unique words as columns of the table and three text examples as rows. In each cell of the table, there is the number of words in the text in question. All words were converted to lower-case before the study, as there would be a distinction between upper- and lower-case letters. The words in the column are listed alphabetically.

Thus, the words in the three different sentences have been converted into numerical data. According to this table, the text expressions in the sentence have been quantified according to their frequency.

During Count Vectorizer, data is not kept as words. For this reason, numerical indices are given to the words seen in the columns of Table 1. As seen in Table 2, the vectorization process of the texts has been completed.


*TFIDF (Term Frequency Inverse Document Frequency) Method:*


TFIDF is the weighting factor that shows the importance of a term in the document. At the same time, TFIDF can be defined as the calculation of how relevant a word in a text string is to the text. It is calculated using the formulas in Equations (1) and (2) [23,24].
(1)$$\mathrm{TFIDF}\left(t,d\right)=T.F.\left(t,d\right)*\mathrm{IDF}\left(t\right)$$
(2)$$\mathrm{IDF}\left(t\right)=\log\left(\frac{n+1}{\mathrm{DF}\left(t\right)+1}\right)+1$$

#### 3.2.2. The Classification (Machine Learning Algorithms)

Classification is creating a machine learning model by learning by the computer in cases where the classes of the past data are known and determining which class the new and unknown data will be found through this model. Machine learning studies consist of two steps called training and testing. First, a learning model is created over the data sets where the intended results are known in the training phase. The test phase passes the data with known results through the model and estimates. The success rate is determined by checking the estimated and actual results [23,25]. If the model’s success is at acceptable levels, using this model, estimation is carried out on data whose results are unknown.

In this paper, among the classification algorithms, Naïve Bayes, LGBM (Light Gradient Boosting Machine), SVM (Support Vector Machine Classification—Support Vector Machine), DTC (Decision Tree Classification—Decision Tree Classification), K-NN (K-Nearest Neighbor)—K-Nearest Neighbors), L.R. (Logistic Regression), RFC (Random Forest Classification) were used.

a)
*Naive Bayes Classification Algorithm:*


Thomas Bayes developed the Naive Bayes algorithm in the 18th century [26]. The mathematical representation of the Naïve Bayes algorithm is given in Equation (3).
(3)$$P\left(A\backslash B\right)=\frac{P(B\backslash A)P(A)}{p(B)}$$

Here, the expression “P(A)” means the probability of occurrence of event A, the expression “P(B)” means the probability of occurrence of event B “, the expression P(B\A)” means the probability of occurrence of event B when event “A” occurs [27].

b)
*Light Gradient Strengthening Machine (LGBM) Classification Algorithm:*


LGBM is an algorithm that creates strong models in data sets where weak learning will occur. It has been published as open-source by Microsoft since 2017 [28]. LGBM has a structure that improves the properties of decision trees in terms of running time and memory usage while maintaining predictive success. The algorithm is successful in big data processing thanks to its histogram-based studies [28]. Furthermore, since LightGBM works in a leaf-based structure, the tree primarily grows horizontally, and the tree depth does not increase much. In this way, excessive learning can be prevented [29].

c)
*Random Forest Classification (RFC) Algorithm:*


It is a formation discovered by Leo Breiman in 2001. However, the method itself was developed by Tin Kam Ho, a statistician at Bell Labs, as an extension of the “bagging” method, which Breiman had also independently developed. Breiman took the idea of “bagging” and applied it to decision trees and named this new ensemble method as “Random Forest. The Random Forest algorithm is used by applying the decision trees algorithm “n” times to increase the prediction success rate [30]. It is used in classification, regression, and feature extraction processes. In the R.F. algorithm, there are two parameters, the number of trees “N” and the value “M”, which determines the number of variables in the nodes. The test condition known as the appropriate cut-off value of the preferred variable for branching is determined by the “gini coefficient”. The Gini Coefficient is calculated as in Equation (4) [31].
(4)$$\mathrm{GINI}(T)=1-\overset{n}{\underset{k=1}{\sum }}{\left(\mathrm{Sk}\right)}^{2}$$

The GINI index is calculated at each node and continues until it reaches zero. When it is zero, branching ends, the tree with the lowest error rate has the highest weight, and the tree with the highest error rate has the lowest weight. Afterward, classes vote according to the weight of the trees, and these votes are collected. The tree structure with the highest votes is determined, and that tree is preferred [31,32]

d)
*K-Nearest Neighbors (K-NN) Classification Algorithm:*


The K Nearest Neighbors algorithm, known as KNN (K-Nearest Neighbors), was developed by T.M. COVER and P.E. It was revealed using the “Nearest Neighbor Decision Rule” created by HART. KNN determines to which classification cluster unclassified data is closest among the classified data [33,34,35,36]

e)
*Support Vector Machines (SVM) Classification Algorithm:*


SVM is a machine learning method using hyperplanes. Apart from other linear methods, it is to determine the gap that provides the separation of hyperplanes so that it is the largest. The distinction is not a single linear equation but a range that can be expressed by many equations [37,38]

The SVM algorithm was created by Vladimir Vapnik and Alexey Chervonenkis in 1963 [39]. This method has been used in many fields, such as chemistry [40], physics [41], biology, and technology. To perform the classification process, SVM determines the optimum hyperplane, that is, the decision plane, which separates the classes from each other [42,43]. In the test phase, the position of the data points to the plane, which will be estimated to be included in which class is examined.

f)
*Decision Tree Classification (DTC) Algorithm:*


The Decision Tree algorithm is a frequently used machine learning method. The Decision Trees algorithm is used for regression and classification problems. It is widely used because it is an algorithm that is cheap to create, easy to interpret, and highly reliable. DTC is a method of iterating the group into groups with a clustering algorithm until all values have the same class label [44]. The decision tree classification method performs the prediction process using the tree structure. In this tree structure, there are decision variables at the nodes of the tree and target values to be predicted at the leaves [44]. The features in the data set are from the nodes in the decision trees, and these nodes answer the node questions as true-false or yes-no and are divided into two. After the data is divided, the influencing feature vectors affecting the features are examined, and the nodes with high success information enter the algorithm to branch [45]. There are problems such as the unstable structure of decision trees, differentiation of the result in the slightest change in the data, and over-learning problems [46]. Decision trees are a machine learning method with statistical and probability structures behind them. The entropy value called the complexity value in statistics, is very important. Entropy is the probability of experiencing unexpected events. In other words, entropy is required to predict possible deviations in the algorithm.

g)
*Logistic Regression (L.R.) Algorithm:*


Logistic regression was introduced by Raymond Pearl and Lowell Reed in 1940 [47]. Logistic regression is used to classify independent variables to examine the probability of a categorical outcome [48]. Equation (5) shows the logistic regression equation.
(5)$$W\left(t\right)=Ω\;\frac{exp\left(\alpha +\beta t\right)}{1+exp\left(\alpha +\beta t\right)}$$

In Equation (5), the expression “Ω” indicates the upper limit of the saturation level of “*W*”, the expression “α” the value of the curve on the “x” axis, and the expression “β” the slope of the curve. With this method, the relationship between the independent variables in terms of probability and the regressions resulting from the regression is determined and calculated [48]. Logistic regression algorithm is frequently used in fields such as health [49], social sciences [50], and political sciences [51].

#### 3.2.3. The Testing Process and Performance Metrics

After the machine learning algorithms complete the training phase, the estimation process is made with the data whose results are unknown. It is difficult to control the accuracy once the predicted effects of the unknown data have been determined. In such cases, to test the success of the models, two-thirds of the training data is trained, and the remaining one-third is tested. In this study, training and testing are done by dividing the training data. Thus, the model is tested with data not included in the training, revealing its success.

a)The confusion matrix is a square matrix used to test the accuracy of predictions of classification models. Many machine learning algorithms are used in this study, and these algorithms are put to the test. The user checks the test results, and the most suitable algorithm is selected for the study. The confusion matrix shows the number of correct and incorrect predictions of the models [52,53]. Table 3 shows the confusion matrix.

In this study, hearing the sounds during the hearing test was expressed as “actually positive situation”, not being heard as “actually negative situation”, “predicted positive situation” for the model to be heard as “predicted negative situation”, and “predicted negative situation” for not being heard.

b)The classification error rate is the data that the model predicts incorrectly. It is calculated with the formula in Equation (6) [54].


(6)
$$\mathrm{Classication}\;\mathrm{Error}=\frac{\mathrm{FP}+\mathrm{FN}}{\mathrm{TP}+\mathrm{TN}+\mathrm{FP}+\mathrm{FN}}$$


c)The classification accuracy rate is the value found by subtracting the prediction error rate from the number 1, which is the total probability value. It can also be calculated with the formula in Equation (7). The accuracy rate is the ratio of the values that the machine learning model predicts correctly [55].


(7)
$$\mathrm{Classication}\;\mathrm{accuracy}=\frac{\mathrm{TP}+\mathrm{TN}}{\mathrm{TP}+\mathrm{TN}+\mathrm{FP}+\mathrm{FN}}$$


d)Recall is the rate at which positive situations are predicted positively by the model. It is calculated as in Equation (8) [56].


(8)
$$\mathrm{Recall}=\frac{\mathrm{TP}}{\mathrm{TP}+\mathrm{FN}}$$


e)Precision shows how many of the data used in the study predicted correctly, and it is calculated as in Equation (9) [57].


(9)
$$\mathrm{Precision}=\frac{\mathrm{TP}}{\mathrm{TP}+\mathrm{FP}}$$


f)Specificity is the ratio of correctly predicted negative true values to all true negative values. It is calculated with the formula in Equation (10) [58].


(10)
$$\mathrm{Specificity}=\frac{\mathrm{TN}}{\mathrm{TN}+\mathrm{FP}}$$


g)F-1 Score is a unit of measure that shows the success of machine learning classification algorithms, also called F-1 Score (F-1 Score) or F-Measure (F-Measure), and is calculated with precision and recall values. It is calculated as in Equation (11) [59].


(11)
$$F-1\;score=\frac{2\;x\;\mathrm{Precision}\;x\;\mathrm{Recall}}{\mathrm{Precision}+\mathrm{Recall}}$$


h)The ROC curve graphically shows the relationship between expressions of sensitivity and specificity for a data test. The ROC curve is obtained by plotting the false positive rates (1-specificity) values versus the sensitivity value. In other words, it is a graphic representation of the relationship between the T.P. value and the F.P. value. The area under the ROC curve is expressed as AUC (Area Under Curve), and it is understood that the classification success of the model increases as it is closer to 1 [60].i)The MCC (Matthews Correlation Coefficient) is a correlation coefficient between actual values and predicted values. The MCC value takes a value between −1 and +1. It is calculated with the formula in Equation (12). It is understood that the classification success of the model increases as the MCC value approaches 1 [61]


(12)
$$\mathrm{MCC}=\frac{\mathrm{TP}\;x\;T.N.\;-\;F.P.\;x\;F.N.}{\sqrt{\left(TP+FP\right)\left(TP+FN\right)\left(TN+FP\right)\left(TN+FN\right)}}$$


j)Log Loss, also called Logistic Regression Loss or Cross-Entropy Loss, takes into account uncertainty in addition to how much the model estimates differ from the true value. The more the estimated estimates deviate from the true values, the higher the Log Loss value [62]

When the prepared dataset is visualized, the P300 signal created on the EEG signals by the given sound stimulus is shown in blue, and the EEG signals obtained in the absence of the sound stimulus as orange, the P300 effect in the data obtained from all experimental studies is shown in Figure 7. As seen in Figure 7, while there is a separation between the signals heard and those not heard in channels 1–13, this separation is seen more clearly in channels 1–8. Therefore, artificial intelligence studies were applied and evaluated for 16 channels, 1–13 and 1–8.

## 4. Results

In this paper, hearing tests and simultaneous EEG recordings were performed in a test environment. In a quiet and calm room, under inconstant and the same conditions (temperature, test stages, test equipment, etc.), nine different people were tested with the Matlab hearing test interface, and the effects of each sound and silence on the brain were recorded with an EEG recorder. Data pre-processing studies were applied to EEG recordings, and then machine learning classification algorithms were tried.

In classical audiometry devices, wavelength and frequency values are given sequentially. With the designed hearing test interface, wavelength and frequency values of sounds can be assigned sequentially and randomly. In addition, hearing tests can be performed separately for the left and right ears and both ears in a sequential manner. In this study, the hearing test was carried out sequentially for both ears, with the wavelength and frequency settings of the sound being random. Figure 8 shows the result of a hearing test performed with the hearing test interface. According to the hearing test report in Figure 8, the sound signal with the lowest amplitude heard at each frequency is marked separately for the left and right ears. The left ear is shown in blue with the symbol “x”, and the right ear with the symbol “o” in red.

The effects of the sounds from the MATLAB hearing test interface on the brain are recorded with a 16-channel EEG recorder. The obtained performance results are shown in Table 4. ROC curves obtained with seven different classification algorithms using 16 channels of EEG signals are given in Figure 9. The obtained other performance results are shown in Table 5.

In the EEG data taken over 16 channels, the P300 effect is seen in channels 1–13. The following results were obtained by running machine learning algorithms for the first 13 channels. The obtained performance results with the first 13 channels are shown in Table 6. ROC curves obtained with seven different classification algorithms using the first 13 channels of EEG signals are given in Figure 10. Finally, the obtained other performance results with the first 13 channels are shown in Table 7.

Effect of P300 on EEG data received over 16 channels 1–13. However, seen on channels 1–8. The separation of heard and unheard sounds in channels is more pronounced. The following results were obtained by running machine learning algorithms for the first eight channels. The obtained performance results with the first eight channels are shown in Table 8. ROC curves obtained with seven different classification algorithms using the first eight channels of EEG signals are given in Figure 11. The obtained other performance results with the first eight channels are shown in Table 9.

When the accuracy rate (accuracy) metric is examined, LGBM was the most successful algorithm with 83% in the study for 16 channels, LGBM was the most successful algorithm with 84% in the study for 13 channels, and the most successful algorithm was LGBM with 82.1% in the study for eight channels. When the precision metric is examined, LGBM was the most successful algorithm with 84.3% in the study for 16 channels, LGBM was the most successful algorithm with 86.2% in the study for 13 channels, and the most successful algorithm was RFC with 87% in the study for eight channels. When the sensitivity (recall) metric is examined, LGBM was the most successful algorithm with 82.7% in the study for 16 channels, LGBM was the most successful algorithm with 81.9% in the study for thirteen channels, and LGBM was the most successful algorithm with 82.5% in the study for 8 channels. When the F1 score metric is examined, LGBM was the most successful algorithm with 83.5% in the study for 16 channels, LGBM was the most successful algorithm with 84% in the study for 13 channels, and LGBM was the most successful algorithm with 82.5% in the study for 8 channels. When the MCC metric is examined, LGBM was the most successful algorithm with a value of 0.66 in the study for 16 channels, LGBM was the most successful algorithm with a value of 0.681 in the study for 13 channels, and LGBM was the most successful algorithm with a value of 0.641 in the study for 8 channels. When the log-loss (Log-loss) metric is examined, LGBM was the most successful algorithm with 0.397 in the study for 16 channels, LGBM was the most successful algorithm with a value of 0.383 in the study for 13 channels, and LGBM was the most successful algorithm with a value of 0.422 in the study for 8 channels. When the AUC ratiometric in the ROC curve is examined, The most successful algorithm was LGBM, with 88% in the study for 16 channels, LGBM was the most successful algorithm with 89.9% in the study for 13 channels, and the most successful algorithm was LGBM with 88.2% in the study for 8 channels.

When the machine learning model success metrics were examined, it was understood that the LGBM algorithm made more successful predictions than other algorithms. It was determined that the success metrics changed when the EEG data of the first 13 channels and the first eight channels, in which the P300 effect was detected, instead of the 16-channel EEG data, were taken for training and tested with machine learning algorithms. Table 10 shows that the training with 13-channel EEG data obtained from the EEG recorder is more successful than the training with 16-channel and 8-channel EEG data.

Audiologists play different sound waves and frequencies to individuals using audiometry equipment and record their responses through button presses. This process allows audiologists to determine which frequency and wavelength of sound are audible to the person being tested. However, human-related factors such as delayed button presses, non-responses, or forgetfulness can lead to inaccurate recordings by audiologists. To mitigate this issue, this project proposes the use of an EEG headband fitted with electrodes that can be worn by the patient. Random sound waves and frequencies will be presented to the patient and the EEG signals detected by the headband will indicate if the patient heard the sound or not. This project aims to make the work of audiologists easier and more accurate, but it is still recommended that final decisions be made by audiologists. To utilize this system, audiologists only need to fit the patient with the EEG headband, input their demographic information into the system, and run the software. The system will then present sounds randomly and record the corresponding EEG signals. These EEG signals will be analyzed using machine learning models to make predictions.

## 5. Conclusions

In this study, the data obtained from the 16-channel EEG signals during the hearing test performed with the MATLAB hearing test interface were analyzed using machine learning classification algorithms with three methods: eight channels, thirteen channels, and sixteen channels. In the study conducted for sixteen channels, the most successful algorithm was LGBM, with a maximum Cross-Validation value of 77.5%. In the study conducted for thirteen channels, it was LGBM with a maximum Cross-Validation value of 78.7%. The study conducted for eight channels was LGBM with a maximum Cross-Validation value of 77.4%. The LGBM algorithm is the most successful in the studies conducted for all channels and channels where the P300 effect is seen, and there was not much difference between success rates. While the random estimation rate of an EEG signal as heard-unheard was 50%, the 78.7% estimation rate was determined by the LGBM algorithm.

In this study, which was carried out with the MATLAB hearing test interface and simultaneously recorded EEG data, it is estimated that the success rates may increase if the data are collected from more people with different demographic (age, gender, etc.) characteristics and working in different occupational groups and estimated by machine learning algorithms. In addition, during the hearing test, the sound isolation of the room where the test is performed and the filtering feature of the external sounds of the headphones used will ensure that the sounds given during the test will be heard more accurately so that the EEG data will be recorded more accurately. Thus the success rates of the machine learning algorithms will be higher. Increasing the number and diversity of volunteers can lead to more successful and consistent prediction results.

In this study, data was collected from nine volunteers. The data collection process from each volunteer takes between 3–4 min. The sounds given to the volunteers were randomly given, so if a sound was not heard, sounds with a smaller wavelength than that sound was not given. Therefore, the data collection process from each volunteer shows differences. The worst case is when each frequency starts from the smallest wavelength and increases. In this case, the data collection process from one volunteer takes about 10 min. The pre-processing process and the creation of the machine learning model for the data of the nine volunteers took about seven minutes. As the number of volunteers increases, the training time will likely increase. The prediction time is about five seconds, regardless of the number of volunteers and the data size. However, this time does not include the time to collect EEG signals from the person. Collecting EEG signals takes about 3–4 min. Time measurements were made on cloud servers with 8-core CPUs and 32 GB of RAM.

## Figures and Tables

**Figure 1 diagnostics-13-00575-f001:**
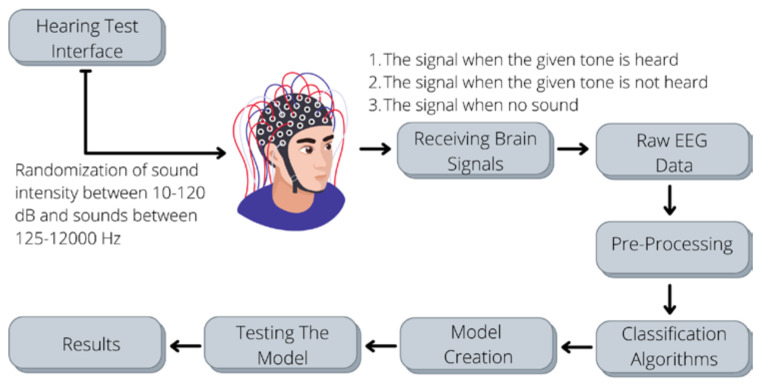
The proposed procedure in our study.

**Figure 2 diagnostics-13-00575-f002:**
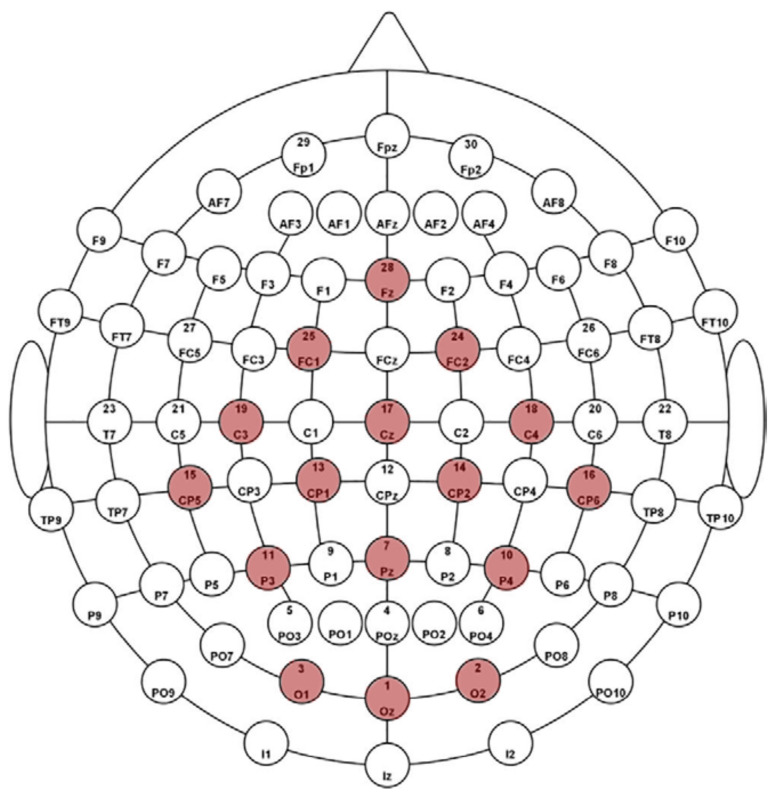
The representation of located EEG electrodes on the scalp [18].

**Figure 3 diagnostics-13-00575-f003:**
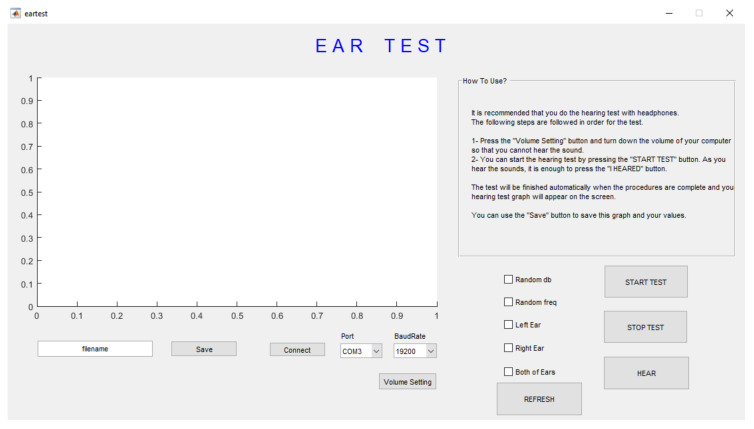
The designed hearing test interface start screen in the study.

**Figure 4 diagnostics-13-00575-f004:**
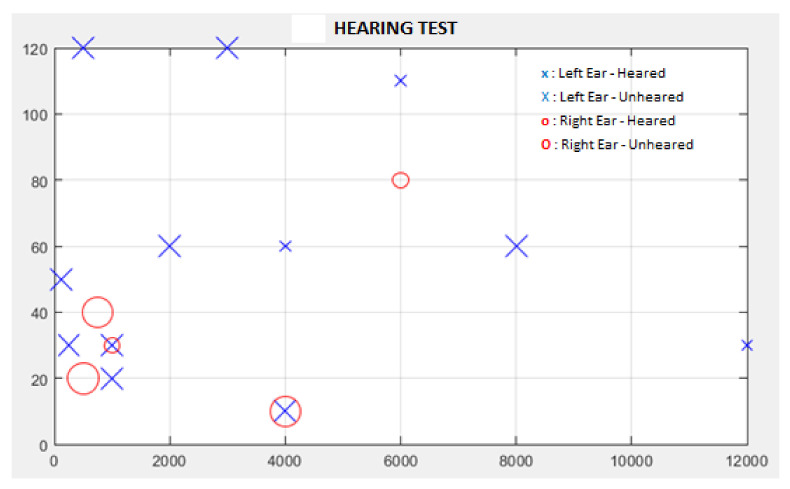
Representative audible and unheard sounds during the test.

**Figure 5 diagnostics-13-00575-f005:**
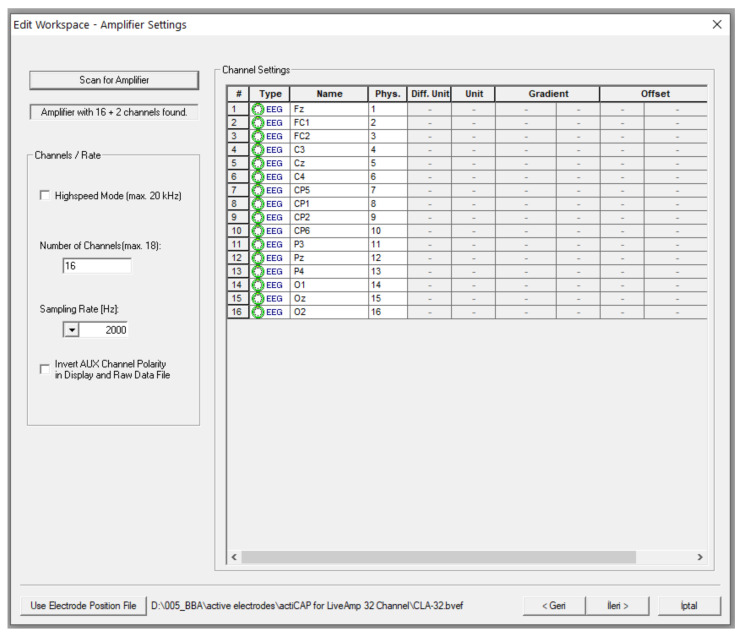
Recorder software settings in the EEG signal analysis.

**Figure 6 diagnostics-13-00575-f006:**
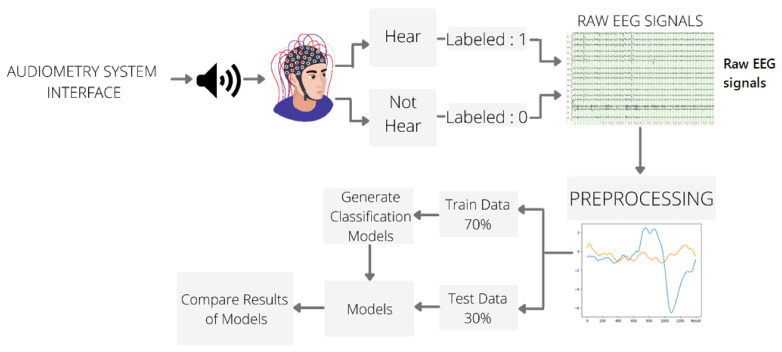
Schematic representation of the proposed automatic audiometric system design based on Machine Learning Methods using the EEG signals.

**Figure 7 diagnostics-13-00575-f007:**
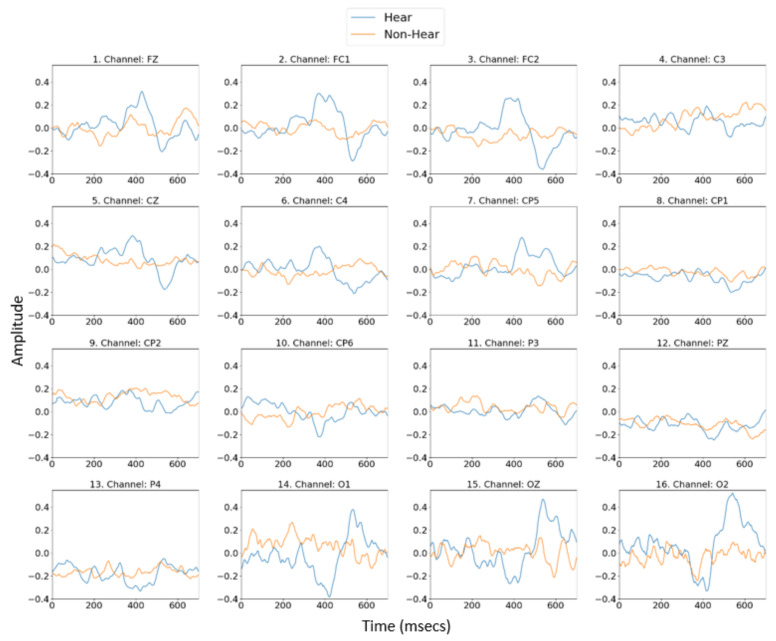
P300 signal effect on the hearing performance in data from all experimental studies.

**Figure 8 diagnostics-13-00575-f008:**
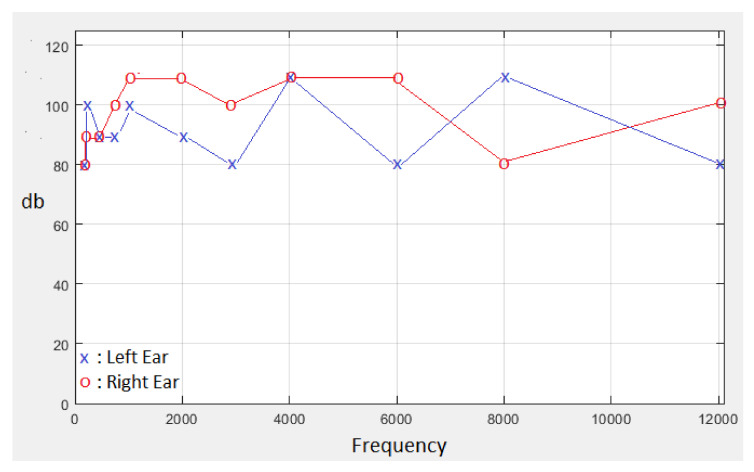
Hearing test results performed with MATLAB GUI hearing test interface.

**Figure 9 diagnostics-13-00575-f009:**
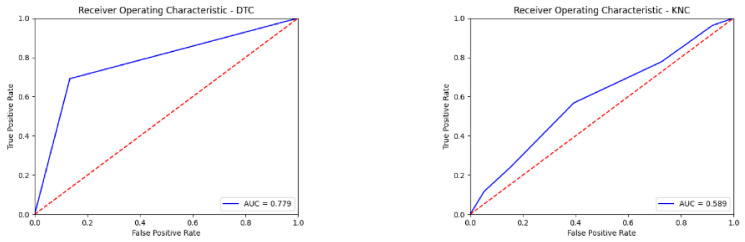
ROC curves obtained with seven different classification algorithms using 16 channels of EEG signals.

**Figure 10 diagnostics-13-00575-f010:**
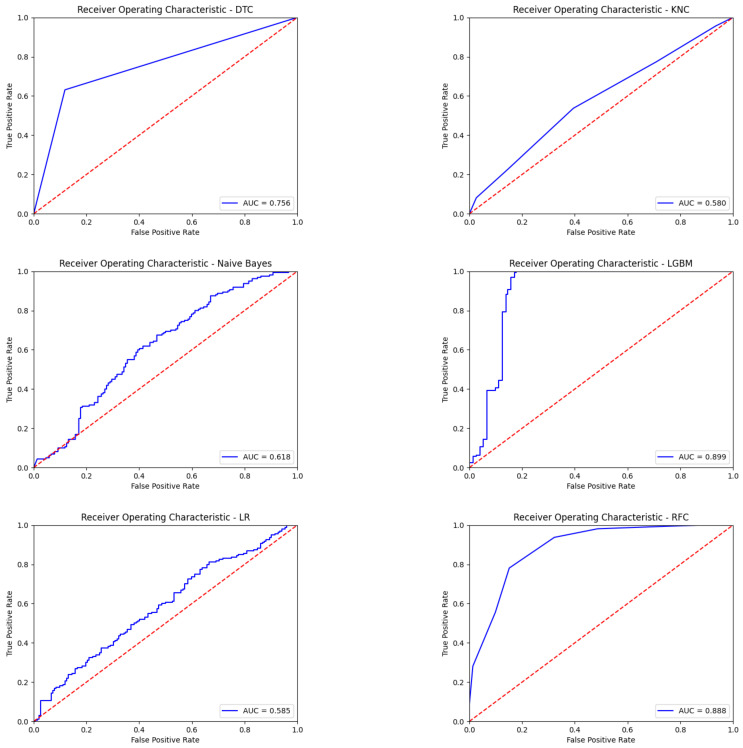
The obtained ROC curves with seven different classification algorithms using the first 13 channels of EEG signals.

**Figure 11 diagnostics-13-00575-f011:**
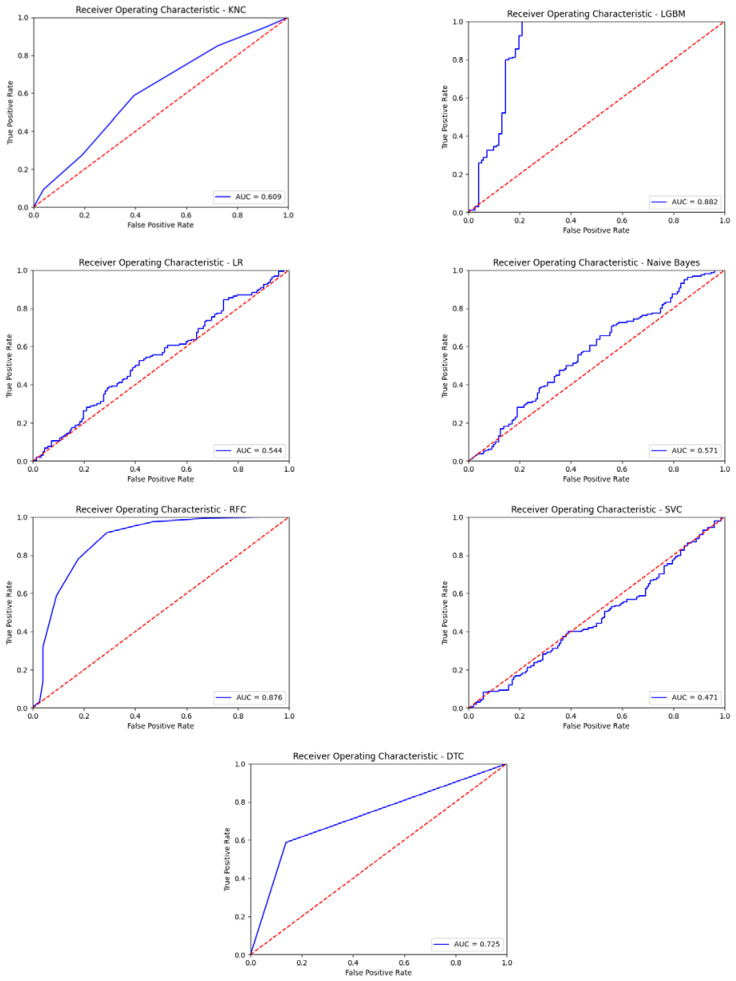
The obtained ROC curves with seven different classification algorithms using the first eight channels EEG signals.

**Table 1 diagnostics-13-00575-t001:** The working of Count Vectorizer.

	A	This	Study	Source	Text	Example	Vector	Time
1. text	0	1	1	0	1	1	0	0
2. text	1	0	0	1	1	2	1	0
3. text	1	1	1	1	2	0	0	1

**Table 2 diagnostics-13-00575-t002:** The completed Count Vectorizer.

0	1	2	3	4	5	6	7
0	1	1	0	1	1	0	0
1	0	0	1	1	2	1	0
1	1	1	1	2	0	0	1

**Table 3 diagnostics-13-00575-t003:** Confusion Matrix.

		Actual		Total
		Positive (1)	Negative (0)	
Predicted	Positive (1)	T.P.	FP	TP + FP
	Negative (0)	FN	TN	FN + TN
Total		TP + FN	FP + TN	TP + FN + FP + TN = N

TP: It refers to positive situations that the model predicts positively. F.P.: It refers to the positive cases that the model predicts negatively. F.N.: Expresses negative situations that the model predicts positively. TN: It refers to negative situations that the model predicts negatively.

**Table 4 diagnostics-13-00575-t004:** The performance results using 16 channels of EEG signals with machine-learning classification algorithms.

Algorithm	TN	FP	FN	TP	Accuracy(%)	MSE(%)	Recall(%)	Precision(%)
N.Bayes	142	8	152	10	48.7	51.3	6.2	55.6
LGBM	125	25	28	134	83.0	17.0	82.7	84.3
SVM (SVC)	150	0	162	0	48.1	51.9	0	-
Decision Tree Classifier (DTC)	130	20	50	112	77.6	22.4	69.1	84.8
k-NN (KNC)	91	59	70	92	58.7	41.3	56.8	60.9
LR	125	25	109	53	57.1	42.9	32.7	67.9
Random Forests Classifier (RFC)	133	17	81	81	68.6	31.4	50.0	82.7

**Table 5 diagnostics-13-00575-t005:** The obtained other performance metrics results using 16 channels of EEG signals with machine-learning classification algorithms.

Algorithm	F1 (%)	Specificity(%)	MCC	Log Loss
N.Bayes	11.1	94.7	0.018	5.965
LGBM	83.5	83.3	0.66	0.397
SVC	-	100	0	0.692
DTC	76.2	86.7	0.56	7.749
KNC	58.8	60.7	0.175	2.191
LR	44.2	83.3	0.185	0.688
RFC	62.3	88.7	0.416	0.519

**Table 6 diagnostics-13-00575-t006:** The performance results using the first 13 channels of EEG signals with machine-learning classification algorithms.

Algorithm	TN	FP	FN	TP	Accuracy(%)	MSE(%)	Recall(%)	Precision(%)
N.Bayes	132	20	139	21	49.0	51.0	13.1	51.2
LGBM	131	21	29	131	84.0	16.0	81.9	86.2
SVC	152	0	160	0	48.7	51.3	0	-
DTC	134	18	59	101	75.3	24.7	63.1	84.9
KNC	92	60	74	86	57.1	42.9	53.8	58.9
LR	109	43	100	60	54.2	45.8	37.5	58.3
RFC	137	15	71	89	72.4	27.6	55.6	85.6

**Table 7 diagnostics-13-00575-t007:** The obtained other performance metrics results using the first 13 channels EEG signals with machine learning classification algorithms.

Algorithm	F1 (%)	Specificity(%)	MCC	Log Loss
N.Bayes	20.9	86.8	0	7.093
LGBM	84.0	86.2	0.681	0.383
SVC	-	100	0	0.693
DTC	72.4	88.2	0.528	8.524
KNC	56.2	60.5	0.143	1.885
LR	45.6	71.7	0.098	0.690
RFC	67.4	90.1	0.485	0.484

**Table 8 diagnostics-13-00575-t008:** The performance results using the first eight channels of EEG signals with machine learning classification algorithms.

Algorithm	TN	FP	FN	TP	Accuracy(%)	MSE(%)	Recall(%)	Precision(%)
N.Bayes	131	21	133	27	50.6	49.4	16.9	56.3
LGBM	124	28	28	132	82.1	17.9	82.5	82.5
SVC	152	0	160	0	48.7	51.3	0	-
DTC	131	21	66	94	72.1	27.9	58.8	81.7
KNC	92	60	66	94	59.6	40.4	58.8	61.0
LR	104	48	97	63	53.5	46.5	39.4	56.8
RFC	138	14	66	94	74.4	25.6	58.8	87.0

**Table 9 diagnostics-13-00575-t009:** The obtained other performance metrics results using the first eight channels EEG signals with machine learning classification algorithms.

Algorithm	F1 (%)	Specificity(%)	MCC	Log Loss
N.Bayes	26.0	86.2	0.042	7.787
LGBM	82.5	81.6	0.641	0.422
SVC	-	100	0	0.693
DTC	68.4	86.2	0.466	9.631
KNC	59.9	60.5	0.193	2.170
LR	46.5	68.4	0.081	0.693
RFC	70.1	90.8	0.520	0.495

**Table 10 diagnostics-13-00575-t010:** The obtained performance metrics results using 16 channels, the first 13 channels, and the first eight channels EEG signals with the LGBM algorithm.

Algorithm	16 Channels EEG	13 Channels EEG	8 Channels EEG
Accuracy (%)	83.0	84.0	82.1
MSE	17.0	16.0	17.9
Recall (%)	82.7	81.9	82.5
Precision (%)	84.3	86.2	82.5
F1 (%)	83.5	84.0	82.5
Specificity (%)	83.3	86.2	81.6
MCC	0.66	0.681	0.641
Log Loss	0.397	0.383	0.422

## Data Availability

We can send the datasets at the request of the authors.

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
