# Peer review of "A Novel Automatic Audiometric System Design Based on Machine Learning Methods Using the Brain’s Electrical Activity Signals"

_diagnostics, 2023, doi:10.3390/diagnostics13030575_

Round 1
Reviewer 1 Report
A novel automatic audiometric system design based on machine learning methods using the brain's electrical activity signals
In their study brain signals were measured with Brain Products Vamp 16 EEG device, and then EEG raw data were created using the Brain Vision Recorder program and MATLAB. After the data set was created from the signal data produced by the heard and unheard sounds in the brain, machine learning processes were carried out with the PYTHON programming language.
The results are very promising, and the application problem has significant importance with respect to big data problem solving.
There are some minor points which need to be improved.
1. The information given in Table 1 should be detailed.
2. The limitations of this study should be given.
3. The introduction should be divided into two and a literature review title.
4. The contributions of the work given in the introduction to this paper can be improved.
5. Please give some examples of the proposed method in the paper.
6. The novelty of the proposed method should be highlighted. The authors should clarify the paper's contributions in the introduction section.
7. In the introduction, the motivation of the paper needs to be articulated far more clearly.
8. There are some spelling mistakes in the article. The article should be read from beginning to end.
Author Response
Reviewer 1:
A novel automatic audiometric system design based on machine learning methods using the brain's electrical activity signals
In their study brain signals were measured with Brain Products Vamp 16 EEG device, and then EEG raw data were created using the Brain Vision Recorder program and MATLAB. After the data set was created from the signal data produced by the heard and unheard sounds in the brain, machine learning processes were carried out with the PYTHON programming language.
The results are very promising, and the application problem has significant importance with respect to big data problem solving.
There are some minor points which need to be improved.
- The information given in Table 1 should be detailed.
Response: Thank you for your valuable comment.
An example of the Count Vectorizer method is given in Table 1. The number of times words appear in the sentence is shown in the table. The word "example" is found once in the 1st text expression and twice in the 2nd text expression. The text in the three sentences has been converted into numerical data; using this table, the expressions in the sentences are quantified based on their frequency.
A similar explanation is added in the relevant section of the article.
- The limitations of this study should be given.
Response: Thank you for your valuable comment.
In this study, data was collected from nine participants. The data collection process for each participant takes between 3-4 minutes. The sounds presented to the participants were randomly chosen, so if a sound is not heard, sounds with a shorter wavelength will not be presented. Therefore, the data collection process for each participant varies. The worst-case scenario is when each frequency starts with the shortest wavelength and increases. In this case, the data collection process for one participant takes around 10 minutes. The preprocessing of the data and creating the machine learning model for the data of the nine participants took around 7 minutes. As the number of participants increases, the training time will also likely increase. The prediction time is around 5 seconds, regardless of the number of participants and the data size. However, this time does not include the time to collect EEG signals from the person, which takes around 3-4 minutes. The time measurements were taken on cloud servers with 8-core CPUs and 32 GB of RAM.
A similar explanation is added in the relevant section of the article.
- The introduction should be divided into two and a literature review title.
Response: Thank you for your valuable comment.
The introduction section was divided into two subheadings, "Introduction" and "Literature Review."
- The contributions of the work given in the introduction to this paper can be improved.
Response: Thank you for your valuable comment.
This study aims to develop an autonomous hearing test using EEG signals and machine learning methods. The test involves using sounds of different amplitudes and wavelengths randomly assigned to the person being tested through a MATLAB interface. The person listens to the sounds through headphones and indicates whether or not they hear them, while EEG signals are recorded to measure brain activity. The EEG data is preprocessed and analyzed using various machine learning algorithms to determine which sounds the person heard and which they did not hear. The best-performing algorithm in the study was the Light Gradient Strengthening Machine (LGBM), which achieved a success rate of 84%. The results suggest that hearing tests can be performed using EEG but may still require evaluation by an audiologist or doctor.
- Please give some examples of the proposed method in the paper.
Response: Thank you for your valuable comment.
In this study, EEG signals were treated as text expressions, and the Count Vectorizer and TF-IDF methods were applied. These approaches are generally used in text classification problems. Priyatan Naravajhula and colleagues have conducted a study on spam classification prediction using text data such as SMS and Twitter.[1]
- Naravajhula, P.; Naravajula, A. Spam Classification: Genetically Optimized Passive-Aggressive Approach. SN Computer Science 2023, 4, 93
- The novelty of the proposed method should be highlighted. The authors should clarify the paper's contributions in the introduction section.
Response: Thank you for your valuable comment.
This study aims to perform hearing tests using brain waves. The response of the brain to sounds given during the hearing test is analyzed to determine whether the sound is heard or not. The detection of brain signal responses to the sounds given with EEG device and converting it into hearing test with AI can be considered an innovative aspect.
A similar explanation is added in the relevant section of the article.
- In the introduction, the motivation of the paper needs to be articulated far more clearly.
Response: Thank you for your valuable comment.
This study aims to convert audiometric tests into an autonomous form. The sounds (with different wavelengths and frequencies) used in standard hearing tests are randomly given to the person via headphones, and the effects of these sounds on the brain are examined. The waves generated in the brain by the sounds are obtained from the EEG device. The EEG data is tested with classification algorithms to determine whether the sounds are heard or not. When the literature is examined, it is seen that EEG signals are used to detect different diseases. Also, different methods have been studied for the hearing test. In this study, the aim is to perform the hearing test based on the data obtained from the EEG device.
- There are some spelling mistakes in the article. The article should be read from beginning to end.
Response: Thank you for your valuable comment.
Our revised paper has been checked and edited by a Native English speaker. In addition, these spelling errors have been corrected.

Reviewer 2 Report
This is potentially an interesting paper, but would require considerable rewriting to make it readable and useful to audiometrists and others who are not machine learning specialists.
Insufficient background information is provided in the Introduction, and the Materials and Methods section focuses on the machine learning algorithms and computing rather than the experimental methods used.
On the other hand, the Introduction contains a lot of details with little relevance to the main arguments of the paper.
How were the machine learning methods used selected?
No mention is made of application for or granting of ethical approval for this study. Information on participant recruitment, numbers and demographics (age, gender, for example) is not provided. Nor is any information provided on the 16, 13, 8 EEG electrode locations used. Why was 1-12 Hz band-pass filtering applied?
Results: ROC curves are provided, but no other statistical analysis.
Conclusions could include a section on how practical it would be for audiometrists to apply the methods described in daily practice. Would they have the time and skills necessary?
Referencing is poor. I have never before read a paper that used 10 Masters Theses as primary references.
Errors of fact: Was Breiman really the originator of the Random Forest method?
Errors of typography: Why are some surnames/family names in capitals and others not, and why are first names given for some researchers and not others?
Figure 1: 'Testin' should be 'Testing'. Other Figures are too small to be easily legible.
I am left with the impression that this study was poorly conducted, inadequately researched, and that results and conclusions are only partially reported.
Author Response
Reviewer 2:
This is potentially an interesting paper, but would require considerable rewriting to make it readable and useful to audiometrists and others who are not machine learning specialists.
Insufficient background information is provided in the Introduction, and the Materials and Methods section focuses on the machine learning algorithms and computing rather than the experimental methods used.
On the other hand, the Introduction contains a lot of details with little relevance to the main arguments of the paper.
- How were the machine learning methods used selected?
Response: Thank you for your valuable comment.
The algorithms selected in our study were preferred because they showed superior performance in ML and succeeded in analyzing EEG signals. Selected classification algorithms also have features of being used online. Naïve Bayes, Light Gradient Strengthening Machine (LGBM), support vector machine (SVM), decision tree, k-NN, logistic regression, and random forest classifier algorithms were used. In the analysis of EEG signals, Light Gradient Strengthening Machine (LGBM) was obtained as the best method.
- No mention is made of application for or granting of ethical approval for this study. Information on participant recruitment, numbers and demographics (age, gender, for example) is not provided. Nor is any information provided on the 16, 13, 8 EEG electrode locations used. Why was 1-12 Hz band-pass filtering applied?
Response: Thank you for your valuable comment.
For this study, data were obtained from nine volunteers with varied characteristics, including 7 males and 2 females. The oldest male volunteer was 44 years old, the youngest male volunteer was 28 years old, and the average age of the male volunteers was 35.14. The oldest female volunteer was 61 years old, the youngest female volunteer was 38 years old, and the average age of the female volunteers was 49.5. The average age of all nine volunteers was calculated to be 38.33. The participants were employed in various occupations. Ethical board approval was obtained before collecting data from the volunteers.
A similar explanation is added in the relevant section of the article.
In this study, a 16-channel EEG device was used. When the data obtained from volunteers is visualized, the P300 wave is detected on the images. In addition to the 16 channels used, 13 channels and 8 channels where the P300 effect is more intense within these 16 channels are also used as training data. The locations of these channels are obtained from different studies.
Fig 1. Proposed design for the 16-channel EEG system with dry sensors. (a) Dry EEG sensor with a 15 mm diameter, a 7 mm depth, and 8 probes. The travel distance of each probe is 3 mm. There is a unique rubber pad around the bottom surface of the sensors. (b) Wireless EEG acquisition system with a preamplifier, an ADC, a microcontroller, and a wireless module. Each circuit board is 36 mm in width. (c) Size-adjustable soft cap with 16 dry EEG sensors. The placement of each sensor is in accordance. (d) Standard 10-20 EEG system. [1]
Fig 2. Positions of the 16 electrodes, including their number and their designations [2]
Before machine learning algorithms, data preprocessing work was carried out. To filter the data, a band-pass filter (BPF) was used to remove noise and unnecessary information from low frequency bands (1 Hz) and unnecessary information from high frequency bands (12 Hz) [3].
- Liao L.; Wu S.; Liou C.; Lu S.; Chen S.; Chen S.; Ko L.; Lin C. A Novel 16-Channel Wireless System for Electroencephalography Measurements With Dry Spring-Loaded Sensors. IEEE Transactıons On Instrumentatıon And Measurement 2014, 63, 1545
- Bhattacharya J. Complexity analysis of spontaneous EEG. Acta Neurobiol. Exp 2000, 60,495-501
- Turnip A.; Hong K. Classıfying Mental Actıvities From Eeg-P300 Signals Using Adaptive Neural Networks. International Journal Of Innovative Computing Information And Control 2012, 8(9), 6429-6443
- Results: ROC curves are provided, but no other statistical analysis.
Response: Thank you for your valuable comment.
Our study used many performance indices other than ROC curves to compare algorithms. These are: TN rate, FP rate, FN rate, TP rate, Accuracy(%), MSE(%), Recall(%) and Precision (%).Not only ROC but also other performance indices were used to select the best ML classification model. Obtained performance results are given in Tables 4,5,6 and 7. And in these tables, all performance criteria are given.
- Conclusions could include a section on how practical it would be for audiometrists to apply the methods described in daily practice. Would they have the time and skills necessary?
Response: Thank you for your valuable comment.
Audiologists use audiometry equipment to administer different sound waves and frequencies to the person being tested. The individual will press a button when they hear the sound, allowing audiologists to record which frequency and wavelength of sound are audible. However, human-related factors such as delayed button press, non-response, or forgetfulness may result in inaccurate recordings by audiologists. In this project, an EEG headband fitted with electrodes is worn by the patient, and random sound waves and frequencies are presented to the patient. The system can detect whether the patient heard the sound based on the EEG signals detected. This project aims to make the work of audiologists easier, but it is still recommended that audiologists make final decisions. It is not expected that audiologists will need to acquire new skills to use such a project.
A similar explanation is added in the relevant section of the article.
- Referencing is poor. I have never before read a paper that used 10 Masters Theses as primary references.
Response: Thank you for your valuable comment.
The following articles are included as references.
- Rajkumar S.; Muttan S.; Sapthagirivasan V.; Jaya V.; Vignesh S.S. Software intelligent system for effective solutions for hearing impaired subjects. Int. J. Med. Inform 2017, 97,152–162
- Fürbass F.; Kural M.A.; Gritsch G.; Hartmann M.; Kluge T.; Beniczky S. An artificial intelligence-based EEG algorithm for detection of epileptiform EEG discharges: Validation against the diagnostic gold standard. Clin. Neurophysiol 2020, 131(6),1174–1179
- Tan S.L.; Loh S.K.; Chee W.C. Speech-enabled pure tone audiometer. Int. Symp. Intell. Signal Process. Commun. Syst. ISPACS 2007 , 361–364
- Fayçal Y.; Wahiba B.; Lotfi B.; Ratiba B.; Benia A. Computer audiometer for hearing testing. Int. Conf. Adv. Electron. Micro-electronics, ENICS 2008, 111–114
- Hossain M.Y.; Doulah A. B. M. S. U. Detection of Motor Imagery (MI) Event in Electroencephalogram (EEG) Signals using Artificial Intelligence Technique. IEEE East-West Des. Test Symp. EWDTS 2020, 0–5
- Sriraam N. EEG based automated detection of auditory loss: A pilot study. Expert Syst. Appl., 39(1), 723–731
- Paulraj M. P.; Subramaniam K.; Bin Yaccob S.; Bin Adom A. H.; Hema C. R. A machine learning approach for distinguishing hearing perception level using auditory evoked potentials. IECBES IEEE Conf. Biomed. Eng. Sci. 2014, 991–996
- Naravajhula, P.; Naravajula, A. Spam Classification: Genetically Optimized Passive-Aggressive Approach. SN Computer Science 2023, 4, 93
- Errors of fact: Was Breiman really the originator of the Random Forest method?
Response: Thank you for your valuable comment.
The Random Forest algorithm was first identified by Leo Breiman in 2001. However, the method itself had been developed earlier by statistician Tin Kam Ho at Bell Labs as an extension of the "bagging" method, which Breiman had also independently developed. Breiman adapted the concept of "bagging" to decision trees and dubbed this new ensemble method "Random Forest." The algorithm is implemented by repeatedly applying the decision tree algorithm "n" times to improve the prediction success rate. It is commonly used in classification, regression, and feature extraction processes.
A similar explanation is added in the relevant section of the article.
- Errors of typography: Why are some surnames/family names in capitals and others not, and why are first names given for some researchers and not others?
Response: Thank you for your valuable comment.
The specified errors have been corrected.
- Figure 1: 'Testin' should be 'Testing'. Other Figures are too small to be easily legible.
Response: Thank you for your valuable comment.
The typo has been corrected.

Round 2
Reviewer 2 Report
Thank you for the minor changes you have made.
English is still poor. For example [line 172], ‘channels’ are not placed on the ‘skull’ (i.e., bone). Electrodes are positioned on the skin.
Lit rev: authors’ names are given inconsistently: e.g. ‘Marwa Gargouri’, ‘N. Sriraam’, ‘Paulraj M P’.
As a reader, I would still like to know how the algorithms used were selected.
For instance, Elman Neural Networks are mentioned in the Literature Review, but their exclusion from the methods finally selected is not justified.
I would also like to know what the inclusion/exclusion criteria were for the study participants.
The authors have not dealt with my request for more information about the electrode locations used. A number is insufficient description. However, they have still NOT provided the standard 10-20 EEG system electrode locations in the paper itself, which is a serious shortcoming.
I see that the authors HAVE now provided a rationale for the use of 1-12 Hz band-pass filtering (although not in the paper itself).
The authors have not responded to my suggestion that a section on how practical it would be for audiometrists to apply the methods described in daily practice could be included in the Conclusions. Furthermore, their comment that 'It is not expected that audiologists will need to acquire new skills to use such a project' could usefully be included in the paper, not just their 'Author Response'.
On the other hand, they have included a new section in the Conclusions that should really be in the Methods section.
I would recommend that the authors ask some colleagues from different departments in their universities to read through the paper and provide feedback on how easy they find it to understand, what they feel is missing, and how they think the study could be improved.
Author Response
- English is still poor. For example [line 172], ‘channels’ are not placed on the ‘skull’ (i.e., bone). Electrodes are positioned on the skin.
Response: Thank you for your valuable comment. We have corrected this sentence in the revised paper.
- Lit rev: authors’ names are given inconsistently: e.g. ‘Marwa Gargouri’, ‘N. Sriraam’, ‘Paulraj M P’.
Response: Thank you for your valuable comment. We have corrected this sentence in the revised paper.
- As a reader, I would still like to know how the algorithms used were selected. For instance, Elman Neural Networks are mentioned in the Literature Review, but their exclusion from the methods finally selected is not justified.
Response: Thank you for your valuable comment.
This study investigated whether hearing tests can be conducted using EEG signals. No study was found in the literature that used EEG signals for hearing tests. In this context, the study is unique. The success rate, study duration, and required equipment resources can be improved using different algorithms and methods. This study aimed to show that hearing tests can be conducted using EEG signals using known and proven classification algorithms.
- I would also like to know what the inclusion/exclusion criteria were for the study participants.
Response: Thank you for your valuable comment.
In this study, a general model was aimed to be created as much as possible. Therefore, no restrictions were made in participant selection. The study was carried out with participants of different ages and genders. If the number of participants increases with new participants with different characteristics such as race, age, and gender, the model's success can increase.
- The authors have not dealt with my request for more information about the electrode locations used. A number is insufficient description. However, they have still NOT provided the standard 10-20 EEG system electrode locations in the paper itself, which is a serious shortcoming.
Response: Thank you for your valuable comment.
A 16-channel EEG device was used in this study. Therefore, a section about the electrode positions of the 16-channel EEG device has been added to the article.
- I see that the authors HAVE now provided a rationale for the use of 1-12 Hz band-pass filtering (although not in the paper itself).
Response: Thank you for your valuable comment.
It has been added to the article.
- The authors have not responded to my suggestion that a section on how practical it would be for audiometrists to apply the methods described in daily practice could be included in the Conclusions. Furthermore, their comment that 'It is not expected that audiologists will need to acquire new skills to use such a project' could usefully be included in the paper, not just their 'Author Response'.
Response: Thank you for your valuable comment.
Additions have been made to the Results section.
- On the other hand, they have included a new section in the Conclusions that should really be in the Methods section.
Response: Thank you for your valuable comment.
The section in question has been moved from the results section to the methods section.
- I would recommend that the authors ask some colleagues from different departments in their universities to read through the paper and provide feedback on how easy they find it to understand, what they feel is missing, and how they think the study could be improved.
Response: Thank you for your valuable comment.
The recommendations were evaluated, and adjustments were made according to the criticisms.
